# New Insights into the Chemical Reactivity of Dry-Cured Fermented Sausages: Focus on Nitrosation, Nitrosylation and Oxidation

**DOI:** 10.3390/foods10040852

**Published:** 2021-04-14

**Authors:** Aline Bonifacie, Philippe Gatellier, Aurélie Promeyrat, Gilles Nassy, Laurent Picgirard, Valérie Scislowski, Véronique Santé-Lhoutellier, Laetitia Théron

**Affiliations:** 1Institut National de Recherche pour l’Agriculture, l’Alimentation et l’Environnement (INRAE), UR370 Qualité des Produits Animaux, F-63122 Saint Genès-Champanelle, France; Aline.Bonifacie@inrae.fr (A.B.); philippe.gatellier@inrae.fr (P.G.); veronique.sante-lhoutellier@inrae.fr (V.S.-L.); 2IFIP—Institut du Porc, 7 Avenue du Général De Gaulle, F-94700 Maisons Alfort, France; 3IFIP—Institut du Porc, La Motte au Vicomte, BP 35104, F-35561 Le Rheu CEDEX, France; aurelie.promeyrat@ifip.asso.fr (A.P.); Gilles.Nassy@ifip.asso.fr (G.N.); 4Association Pour le Développement de l’Industrie de la Viande (ADIV), 10, Rue Jacqueline Auriol, F-63039 Clermont-Ferrand, France; laurent.picgirard@adiv.fr (L.P.); Valerie.Scislowski@adiv.ar (V.S.)

**Keywords:** nitrite, nitrate, cured meat, dry-cured fermented sausage, nitrosamine, nitrosylheme, nitrosothiol, oxidation

## Abstract

Nitrite and nitrate are added to cured meat for their bacteriological, technological and sensorial properties. However, they are suspected to be involved in the formation of nitroso compounds (NOCs), such as potentially mutagenic nitrosamines, nitrosylheme and nitrosothiols. Controlling the sanitary and sensorial qualities of cured meat products by reducing these additives requires elucidating the mechanisms involved in the formation of NOCs. To this end, we studied the dose-response relationship of added sodium nitrite and/or sodium nitrate (0/0, 80/80, 0/200, and 120/120 ppm) on the formation of NOCs in dry cured fermented sausages. The results showed a basal heme iron nitrosylation in the absence of NaNO_2_/NaNO_3_ due to starter cultures. This reaction was promoted by the addition of NaNO_2_/NaNO_3_ in the other conditions. Reducing the dose to 80/80 ppm still limits lipid oxidation without the formation of non-volatile nitrosamines. Conversely, the addition of NO_2_/NO_3_ slightly increases protein oxidation through higher carbonyl content. The use of 80/80 ppm could be a means of reducing these additives in dry-cured fermented meat products.

## 1. Introduction

Dry-cured fermented sausages are one of the main cured meat products consumed in France [1]. The combined use of nitrite and/or nitrate is characteristic of such fermented products. The addition of nitrate, knowing that it will be reduced to nitrite, is a traditional curing method [2]. Nitrite and nitrate are added in meat products for their bacteriological and organoleptic properties. Indeed, nitrite prevents the growth of pathogenic bacteria, such as strains of *Salmonella* typhimurium and *Clostridium botulinum* [3,4]. In addition, nitrite is used during curing processes for its antioxidant capacity, promoting product conservation [5]. The antioxidant properties of nitrite can also prevent the formation of aldehydes, some of which are mutagenic, due to lipid peroxidation during the curing process [6]. Moreover, nitrite gives cured meat products their characteristic color and flavor [2,3,7].

Nevertheless, besides these positive effects, nitrite and nitrate are involved in the formation of various nitroso compounds (NOCs), and so their use is controversial. Indeed in oxidative and low pH conditions, nitrite can react with compounds such as secondary amines (through N-nitrosation reaction) to form nitrosamines, some of which are mutagenic [8]. This mutagenicity is due to their degradation into reactive compounds which can cause DNA adducts [9]. In reducing medium, nitrite can also react with the heme iron from myoglobin to form nitrosylheme (through nitrosylation reaction), which gives cured meat its red color [7]. During digestion, nitrosylheme can release NO which could participate in the formation of nitrosamines [10], but this is still a subject of debate [11]. Finally, nitrite can react with free thiol groups (through S-nitrosation reaction) to form nitrosothiols [10]. Nitrosothiols are not dangerous by themselves, but upon digestion they can transfer their NO groups to heme iron and to secondary amines to form nitrosylheme and nitrosamines, respectively [10].

This chemical reactivity, namely nitrosation, nitrosylation and oxidation reactions, can be impacted by the specific technological process used in dry cured fermented meat products. Indeed, these products need several weeks of drying-ageing, according to sausage size and quality schemes [12]. The early stage of the drying-ageing step (fermentation stage) allows the development of the specific technological starter added during batter elaboration. These microorganisms play a role in the sanitary quality and conservation of dried products [2] and have a nitrate-reducing activity [13,14]. Moreover, the drying process reduces water activity (aw) and is accompanied by a reduction of pH [2,15] while, proteolysis and lipolysis occur during aging. Proteolysis and lipolysis are the major biochemical reactions leading to the formation of aroma and flavor precursors in fermented meats [16]. Proteolysis is also implicated in the S- and N-nitrosation process by promoting the release of peptides and free amino acids. Nitrosation in large-size proteins remains very low due to steric hindrance and potential denitrosation by certain surrounding residues [17].

Therefore, it is of great importance to take into account this specificity to reduce the use of additives, knowing that in France, the maximum authorized doses are 120 ppm of sodium nitrite plus 120 ppm of sodium nitrate or 200 ppm of sodium nitrate for products dried during more than 3 weeks [12]. Thus, the aim of this study is to elucidate the reactions and interactions implicated in the formation of NOCs and the oxidation of lipids and proteins in dry-cured fermented meat products, with regard to adding doses of nitrite/nitrate. The effect of modulating added nitrite and/or nitrate concentrations in dry cured fermented sausages is explored. The chemical reactivity of precursors and catalysts in the formation of nitroso compounds and lipid and protein oxidation products is evaluated.

## 2. Materials and Methods

### 2.1. Reagents

All the reagents, i.e., acetone, ascorbate, ethanol, ferrozine, hydrochloric acid, thiobarbituric acid, and the Griess reagent kit for nitrite and nitrate assays (ref: 23479-1KT-F) were purchased from Sigma Aldrich (Saint Louis, MO, USA). The Vivaspin^®^ 2 system (PES membrane with a cut-off of 5 kDa) (ref: VS0212) and the syringe filters with 0.22 µm regenerated cellulose membranes (ref: 17761) were purchased from Sartorius (Göttingen, Germany).

### 2.2. Preparation of Dry-Cured Fermented Meat Samples

The elaboration process is summarized in Figure 1. Dry-cured fermented sausages were made from mixing pieces of porcine shoulders (obtained from Tradival, Lapalisse, France) and back fat at the Technical Institute ADIV (Clermont-Ferrand, France). The shoulders were defatted, skinned, denerved and deboned. After mixing and grinding, commercial starter cultures and ingredients were added (Table 1). The starter culture (Lallemand, France) was a mix of *Lactobacillus sakei*, *Staphylococcus carnosus* and *Staphylococcus xylosus*.

Current industrial formulations include sodium nitrate alone (expressed as NaNO_3_) or a mixture of sodium nitrite and sodium nitrate (expressed as NaNO_2_ and NaNO_3_ respectively). In the following publication, 0 NO_2_/NO_3_ corresponded to a control without sodium nitrite/sodium nitrate added. The added dose of 80 ppm of NaNO_2_ and 80 ppm of NaNO_3_ (80 NO_2_/NO_3_) corresponded to a reasonably safe rate at the sanitary level [3] and is the short-term objective for industrial production. The addition of 120 ppm of NaNO_2_ and 120 ppm NaNO_3_ (120 NO_2_/NO_3_) and 200 ppm of NaNO_3_ (200 NO_3_) corresponded to the maximum authorized by the French Code of Practice [12] for products with nitrite and nitrate, respectively. For each condition studied, the concentration in equivalent mM of total NO added is presented in Table 2 for comparison with data from the literature. Moreover, the concentrations of nitrite, nitrate, nitrosothiols and nitrosamines expressed in ppm are also expressed in µM in Appendix A.

After being stuffed into natural pork casings with a diameter of between 55 and 60 mm and having previously undergone desalting, acid treatment and rinsing. Fresh sausages were dipped into a commercial *Penicillium nalgiovensis* spore solution (BiovitecTexel, Dupont Danisco, France) and hung vertically in a temperature-controlled incubator to carry on the ripening process. The temperature was increased to 24 °C within 6 h, then kept stable for one day and a half. Then, during the drying-ageing period, the temperature was maintained at 13 °C until the dry sausages reached 41% weight loss, corresponding to four weeks. The samples were frozen and stored at −80 °C until analysis.

### 2.3. Biochemical Characterization

#### 2.3.1. Determination of Nitrite and Nitrate Content

Nitrite and nitrate were extracted according to the method of Bonifacie et al. [18]. Briefly, the deprotenization of the extract was performed by centrifugation with a Vivaspin^®^ 2 system with a 5 kDa cut-off threshold (16 °C, 900× *g* for 75 min with an SL 40R centrifuge from Thermo Scientific, Waltham, MA, USA) and the filtrates were used for assays.

Nitrite and nitrate ion contents were then determined using the Griess reaction with a Sigma-Aldrich colorimetric assay kit. Residual nitrite and nitrate were expressed in mg/kg of final product (ppm).

#### 2.3.2. Determination of Nitrosothiol Content

Nitrosothiol contents were determined according to Bonifacie et al. [18]. Briefly, filtrates prepared in Section 2.3.1 were saturated with HgCl_2_ to specifically cleave the S-NO bonds of nitrosothiols [19]. Nitrite ion contents were measured using the Griess reaction and the difference between this measure and the level of nitrite ions initially present in the sample (Section 2.3.1) gave the nitrosothiol content [19]. Nitrosothiols were expressed in mg S-NO/kg of final product (ppm).

#### 2.3.3. Determination of Non-Volatile Nitrosamine Content

Non-volatile nitrosamine content was determined by the method previously described by Bonifacie et al. [18]. Filtrates prepared in Section 2.3.1 were irradiated with a UV lamp, leading to the cleavage of both S-NO bonds of nitrosothiols and N-NO bonds of nitrosamines into nitrite and nitrate.

This measure gave the sum of nitrosothiol and non-volatile nitrosamine contents. The subtraction of nitrosothiol content, obtained as described in Section 2.3.2, gave the non-volatile nitrosamines that were expressed in mg N-NO/kg of meat (ppm).

#### 2.3.4. Evaluation of Free Iron Content

The free iron content was measured by the ferrozine assay according to Stolze et al. [20], with slight modifications to adapt to the cured meat samples [21]. The results were expressed in µM of total free iron content.

#### 2.3.5. Determination of Heme Iron Nitrosylation

Nitrosylheme and the total heme iron content were estimated according to Hornsey [22]. The nitrosylation of heme iron was expressed as the percentage of nitrosylheme to total heme iron.

#### 2.3.6. Lipid Oxidation Measurement

Lipid extraction was performed using the procedure of Folch et al. [23] with slight modifications according to Bonifacie et al. [21]. Lipid oxidation was measured by the thiobarbituric acid reactive substances (TBARS) method according to Mercier et al. [24]. The results of lipid oxidation were then expressed as µg of malondialdehyde (MDA) equivalent per g of lipids.

#### 2.3.7. Protein Oxidation Measurement

Protein oxidation was measured by estimating the carbonyl groups using the method of Oliver et al. [25] with slight modifications [24]. Carbonyl groups were detected by reactivity with 2,4 dinitrophenylhydrazine (DNPH) to form protein hydrazones. Absorbance was measured on a Multiskan spectrum from Thermo Scientific (Waltham, MA, USA). The results were expressed as nanomoles of DNPH fixed per milligram of protein.

Protein oxidation was also evaluated by the free thiol content. Free thiols were measured by a modification of Ellman’s method using 2,2′-dithiobis (5-nitropyridine) (DTNP) [26]. The results were expressed as nanomoles of DTNP bound per milligram of protein.

#### 2.3.8. Proteolysis Index Measurement

The proteolysis index was estimated by the fluorescamine method according to Harkouss et al. [27]. This method is based on specific fluorescamine labeling of N-terminal α-amino groups of peptides and free amino acids. The fluorescence measurements were performed on a Jasco FP-8300 spectrofluorometer, (Jasco, Oklahoma City, OK, USA). The index of proteolysis was expressed in µmoles of peptides equivalent glycine by protein concentration in mg.

### 2.4. Statistical Analysis

The statistical analysis was carried out with STATISTICA software (version 13.3) from TIBCO Software Inc. (Palo Alto, CA, USA). The values for each experimental condition were reported as the mean ± standard error of the mean (SEM) of six independent repetitions. The effect of nitrite/nitrate treatment on the dry-cured fermented sausages was assessed by a one-way analysis of variance (ANOVA): (0 NO_2_/NO_3_; 80 NO_2_/NO_3_; 200 NO_3_; 120 NO_2_/NO_3_). The correlation matrix and ANOVA was performed using a confidence level of 1%, and the post hoc test used was the Tukey test.

## 3. Results and Discussion

The modulation of the dose-response relationship of sodium nitrite and/or sodium nitrate added in dry-cured fermented sausages was evaluated through the formation of nitroso compounds and their precursors and catalysts. The chemical reactivity of NOCs in dry cured fermented sausages will also be discussed.

### 3.1. Impact of Sodium Nitrite/Nitrate Added to Residual Nitrite and Nitrate

The residual nitrite and nitrate level is a key contributor to the chemical reactivity of NOCs because of their role as precursors and their chemical conversion capability. Overall, with no addition of NO_2_/NO_3_, less than 5 ppm of residual nitrate and 0.2 ppm of nitrite was observed (Figure 2). The level of nitrate in 0 NO_2_/NO_3_ condition is due to the presence of natural residual nitrate in meat, from 10 and 15 ppm in fresh pork meat [28]. Moreover, protein oxidation during the curing process can lead to the production of ammonium that is converted into nitrite by ammonia-oxidizing bacteria and then to nitrate by nitrite-oxidizing bacteria [29,30,31].

Then, increasing the added dose led to a significant increase of residual nitrite and nitrate (*p* < 0.01). The addition of 80 ppm of sodium nitrite and 80 ppm sodium nitrate, corresponding to a total addition of 3 mM of NO significantly increased (*p* < 0.01) the level of residual nitrite but not the residual nitrate. Residual nitrite was observed even when only nitrate was added (200 ppm NO_3_, corresponding to 3.2 mM of added NO). The partial reduction of nitrate into nitrite, done by bacterial flora like Staphylococcus carnosum and Staphylococcus xylosus, used in the fermentation process, explains the nitrite content when only nitrate was added [2,3]. Bacterial flora, naturally present in meat or through the addition of microorganisms as starter cultures, express nitrate-reducing activity [13,14]. However, the process of reducing nitrates takes time and this conversion depends on the concentration of the reactants [32], among other things. For the same quantity of nitrate added initially, Christieans et al. [3] reported a similar residual nitrite content in French dry-cured fermented sausages after 34 days storage.

The 120 NO_2_/NO_3_ condition, corresponding to the maximum NO added in our study (4.5 mM), led to 21 ppm residual nitrate and 8 ppm residual nitrite. Sallan et al. [33] reported a residual amount of nitrite in dry fermented sausages of 12 ppm (residual nitrate was not evaluated) for 150 ppm nitrite added initially, i.e., 3.3 mM equivalent of added NO. This value is half that observed in our samples because the sum of the NO added was higher. In conclusion, regardless of whether NO was added through NO_2_ and/or NO_3_, our results demonstrated a constant increase of residual nitrate and nitrite with the molar concentration of added NO.

### 3.2. Impact of Sodium Nitrite/Nitrate Added to Free and Heme Iron

Total free iron (Figure 3A) is implicated in the chemistry of NOCs by catalyzing their formation and that of oxidation products [34]. Total heme iron and nitrosylated-heme iron were quantified and the percentage of nitrosylation was calculated (Figure 3B). We must mention that in the dry-cured fermented sausages, the sum of free and heme iron content led to approximately the same values, i.e., 0.25 mM, for each condition studied, whatever the concentration of NO added.

The free iron level was significantly (*p* < 0.01) higher in the absence of nitrite/nitrate. This result can be explained by a higher release of iron from myoglobin. Moreover, the oxidative process and free radical chemistry can affect the level of free iron. Indeed, nitric oxide can neutralize superoxide radicals by the formation of peroxynitrite (ONOO^−^), thereby preventing the formation of H_2_O_2_ involved in heme iron release [34]. The dose of added nitrite/nitrate had no effect on the total free iron content.

In the 0 NO_2_/NO_3_ condition, the content of total heme iron was significantly (*p* < 0.01) lower than that observed in samples with added nitrite/nitrate, regardless of the concentration. The protective effect of nitrite against the oxidative process involving the heme iron released might explain this result, as mentioned above. We hypothesize that this result was because nitrosylheme is more stable than heme iron, already described in cooked ham [21]. This hypothesis was confirmed by a negative correlation (−0.67) between nitrosylheme and free iron (*p* < 0.001) (Appendix A). Indeed, during oxidative stress, the release of iron from the porphyrin ring is inhibited by its nitrosylation [35]. However, when comparing curing/drying and curing/cooking processes, 35% less iron is released from myoglobin in the drying process than in the cooking one [21]. In dry products, nitrosomyoglobin stays in its native form, while in cooked products, heat modifies the protein part of the heme pocket leading to a nitroso-hemochrome. Myoglobin denaturation under heating may reduce the stability of heme iron [36].

The basal nitrosylation of 21% observed in the 0 NO_2_/NO_3_ condition may be due to the residual nitrate present in meat that was partially reduced into nitrite during sausage aging [31], as described in Section 3.1. This basal rate of nitrosylation shows the importance of studying the residual contents of nitrite/nitrate in products in which they are not added through sodium NO_2_/NO_3_. In addition, this result supports the hypothesis of basal nitrosylation due to the nitrate reductase activity of the bacterial flora in the dry cured fermented sausages.

Adding nitrite/nitrate significantly increased (*p* < 0.01) the percentage of nitrosylation up to 74% for the 120 NO_2_/NO_3_ condition, corresponding to 4.5 mM equivalent of added NO. A significant increase in nitroso-myoglobin was also observed with the addition of NO_2_/NO_3_ in dry cured loins [37] and a percentage of nitrosylation of 66% was observed in traditional Spanish dry fermented sausages with 4.35 mM equivalent of NO added [38].

In conclusion, starter cultures and their nitrate reductase activity can lead to the formation of nitrosylheme even without added sodium NO_2_/NO_3_. Moreover, the addition of sodium nitrite/nitrate reduces the oxidative process leading to the release of iron from myoglobin.

### 3.3. Impact of Sodium Nitrite/Nitrate Added on Nitrosation Reactions

The products of nitrosation, i.e., nitrosothiols and nitrosamines were measured in dry-cured fermented sausages (Figure 4). The nitrosothiol level did not differ significantly, whatever the dose of sodium nitrite/nitrate. No nitrosamine was detected at 0 or 80 NO_2_/NO_3_; they were detected only for the two highest conditions of added sodium nitrite/nitrate, i.e., 120 NO_2_/NO_3_ and 200 NO_3_.

The literature on nitrosation in dry-cured fermented sausages is scarce and, to our knowledge, there are no studies showing the impact of nitrite and nitrate levels on the amount of nitrosothiols and total non-volatile nitrosamines in these products. Nonetheless, one study quantified the levels of five individual non-volatile nitrosamines in salami [39]. The sum of these five nitrosamines was evaluated at 2 ppm, much lower than our results, but the total non-volatile nitrosamine level was not considered. In addition, the aging duration of these products was not reported; a shorter curing time may explain the lower rate of nitrosamines formed compared to those we observed. Indeed, the proteolysis reaction occurring during the curing process promotes the cleavage of peptidyl chains by proteases, thus possibly promoting protein nitrosation. Moreover, the levels of nitrite and/or nitrate added in Danish salami were not specified, so it is difficult to conclude without more information.

The absence of ascorbate added in dry-cured fermented sausages can also explain our results. Indeed, no non-volatile nitrosamines were detected in a cooked meat model with ascorbate, regardless of the concentration of added nitrite [21]. In this product, the addition of ascorbate favored the formation of nitric oxide (NO°), which is a less effective nitrosating agent than nitrosonium ions (NO^+^) formed in low pH conditions and in the absence of a reducing agent such ascorbate. Moreover, in this cooked product, the highest concentration of added sodium nitrite was lower (1.9 mM) than in dry-cured fermented sausages (4.5 mM).

In conclusion, the formation of non-volatiles nitrosamines was modulated by increasing the sodium NO_2_/NO_3_ concentration. Indeed, these nitroso-compounds were formed only above a concentration of 3 mM of total NO added (80 NO_2_/NO_3_).

### 3.4. Impact of Sodium Nitrite/Nitrate Added to Lipid and Protein Oxidation

Lipid oxidation was evaluated in dry-cured fermented sausages by the TBARS test (Table 3) which measures the level of aldehydes, final oxidation products, with among others dialdehydes and 4-hydroxy-2-nonenal (HNE) (Figure 5, Reaction (3)). Values ranged from 6 to 13 µg of MDA equivalent per g of lipids. Lipid oxidation was significantly (*p* < 0.01) lower with added sodium NO_2_/NO_3_, whatever the dose, confirming the powerful antioxidant capability of nitrite. These results are in line with those reported in dry-cured loins with several added doses of nitrite/nitrate, from 0 to 150 ppm [37]. Moreover, as described in Section 3.2., in the absence of added sodium NO_2_/NO_3_, the free iron concentration was higher. Free iron is known to promote lipid oxidation by the Fenton reaction, leading to the formation of free radicals [34]. These mechanisms were strengthened by the positive correlation (0.68) (*p* < 0.001) between the free iron and TBARS (Appendix A). Moreover, nitric oxide ends the oxidation reactions by reacting with lipoperoxide radicals L(O)O° to form peroxynitrite derivatives L(O)ONO and more stable lipid nitro-compound derivatives LONO_2_ [40] (Reaction (1), Figure 5). This reaction prevented the formation of aldehydes (Reaction (2), Figure 5) involved in reaction with TBA (Reaction (3), Figure 5).

In dry-cured fermented sausages, increasing the dose of sodium NO_2_/NO_3_ did not significantly affect the level of lipid oxidation, demonstrating that the lower dose, 80 ppm of NaNO_2_ and 80 ppm of NaNO_3_, was enough to limit lipid oxidation. In dry cured-loins [37], where the process is different from dry-cured fermented sausage, a lower dose, i.e., 37.5 ppm of NaNO_2_ and 37.5 ppm of KNO_3_, was also sufficient to limit lipid oxidation and there was no effect of increasing nitrite/nitrate dose. The decrease of lipid oxidation with added nitrite, regardless of the concentration, has also been described in a cured and cooked product [21], so it appears that the involvement of free iron in the oxidation mechanisms and the antioxidant capacity of nitrite previously described thus occur independently of the curing process.

Carbonyl groups and free thiol content (Table 3) were evaluated in dry-cured fermented sausages as chemical markers of protein oxidation.

Contrary to lipid oxidation, the amount of carbonyls was slightly higher in the presence of added sodium NO_2_/NO_3_, approximately 10%, and this nitrite/nitrate effect was significant (*p* < 0.01). In fermented sausages, Villaverde et al. [41] observed the same increase of protein carbonyl groups, with 75 and 150 ppm of added nitrite (1.6 and 3.3 mM equivalent of total added NO, respectively). This effect tended to disappear for longer periods of conservation, longer than 50 days, but no explanation was provided.

Interactions between the oxidation of lipids and proteins [42] could explain these contrasting results between TBARS and carbonyls.

Firstly, nitrite inhibits lipid oxidation and so decreases the TBARS level by reacting with lipoperoxides, as described in reaction (1) (Figure 5). Nitrite can also interact with NH/NH_2_ groups of basic amino acids and so interferes with protein carbonylation. Indeed, protein carbonyl groups can be generated directly by amino acid oxidation (Reaction (4), Figure 5) [43]. Nevertheless, nitrite has greater efficacy in trapping intermediates in lipid oxidation than in protein carbonylation. Indeed, at neutral pH, the reaction of nitrite with lipoperoxide radicals exhibits a constant rate, k = 2.109 M^−^^1^s^−^^1^ [34], well above that corresponding to amine nitrosation, k of the order of 105 to 106 M^−^^1^s^−^^1^ [44].

Secondly, the formation of Schiff bases can decrease the level of TBARS and simultaneously increase protein carbonyls (when dialdehydes are involved), as shown in reactions (5) and (6) (Figure 5). Indeed, proteins can be carbonylated indirectly by the formation of Schiff bases between free amino groups of proteins (a-terminal amino groups or e-lysine amino groups) and aldehydes (Reaction (5), Figure 5). Dialdehydes like MDA can also react with free amino groups of proteins, leading to the formation of Schiff bases (Reaction (5), Figure 5) and also to a carbonyl group with its second free aldehyde function (Reaction (6), Figure 5) [43]. The mechanisms by which nitrite could favor the formation of Schiff bases is not clear but has already been observed in fermented sausages by Villaverde et al. [41]. Protein conformational changes, due to the nitrosation (nitrosotryptophan) or nitration (nitrotryptophan and nitrotyrosine) of certain amino acids could render amino groups of proteins or peptides more accessible to hydrophobic aldehydes and so favor Schiff bases formation.

Finally, the contrasting impact of nitrite on TBARS and carbonyls could be linked to greater sensitivity of proteins than lipids to highly oxidative peroxynitrite (ONOO^−^) (Reactions (7) and (8), Figure 5) which can be formed from nitrite in the presence of iron following reactions (NO_2_^−^ + H^+^ ↔ HNO_2_) then (Fe^2+^ + HNO_2_ ↔ Fe^3+^ + NO°+ OH^−^) and (NO° + O_2_°^−^ → ONOO^−^), (Reaction (9), Figure 5), and which is an excellent protein oxidant [45].

The free thiol content ranges from 26 to 33 nmol/mg protein, which corresponds to the values found in dry fermented sausages after 29 days [15], with a content up to 29 nmol/mg protein. Overall, there was no effect of sodium NO_2_/NO_3_ on the content of free thiol groups. In meat products, the level of free thiols depends on the oxidation of thiol groups, their reduction and, in the presence of a nitrite source, to their nitrosation/nitrosylation leading to nitrosothiol compounds. For example, free thiols of cysteine can be oxidized in disulfide bonds according to this reaction (2 R-SH → RS-SR), inducing a decrease of free thiols. However, this reaction can be reversed by the reduction of certain disulfide bonds by thioredoxin reductase, the main protein disulfide reductase [46]. The participation of bacterial flora in this reduction could also be considered. The presence of a nitrite source can lead to the formation of nitrosothiols by the addition of NO to free thiols following the reaction (R-SH+NO → R-SNO), as discussed in Section 3.3.

As several mechanisms occur at the same time, it is difficult to assess the share and impact of each of them in the evolution of free thiol content, a chemical marker of protein oxidation that could complement the protein oxidation results initially provided by the carbonyl content.

To conclude, lipid oxidation decreased significantly with the addition of 80 ppm sodium NO_2_/NO_3_, in particular due to the role of the latter in the formation of stable lipid nitro-compound derivatives and in the release of free iron involved in the Fenton reaction. Conversely, this addition leads to a slight increase in the amount of carbonyl groups.

### 3.5. Impact of Added Sodium Nitrite/Nitrate on the Proteolysis Index

It is important to take the proteolysis index into account in the formation of NOCs because of the release of peptides and free amino acids that might be further involved in the chemical reactivity. The proteolysis index was evaluated in dry-cured fermented sausages (Figure 6) and was significantly decreased (*p* < 0.01) by the addition of sodium NO_2_/NO_3_. This effect could be explained by the binding of NO to certain amino acids, thereby modifying the conformation of proteins and inhibiting optimal recognition by proteases [47]. This decrease could also be explained by the adduction of NO at the active site of the proteases, which would make them less effective. Indeed, in meat, cysteine proteases such as cathepsin are highly involved in proteolysis at pH less than or equal to 5. As these proteases act with a cysteine group at their active site, they can lose their activity by the nitrosation of the thiol group, as already described in another model [48]. In addition, the formation of carbonyls in dry-cured fermented sausages, presented in a previous section, could also lead to difficult recognition of proteins by proteases.

The maximum value of 3.5 µmoles of peptides/mg of proteins was reached in the absence of added nitrite/nitrate, matching approximately 26% of proteolysis. The proteolysis index of 1.9 µmoles of peptides in fresh meat was significantly (*p* < 0.01) lower compared to dry sausages at the end of the process in 200 NO_3_ and 120 NO_2_/NO_3_ conditions. The curing process promotes proteolysis as several processing conditions are involved. Indeed, meat peptides generated by the action of endogenous proteases during the ripening of dry-cured fermented sausages have been reported [49]. The addition of starter cultures is also involved in proteolysis due to their acidification capacity [49], as a decrease of pH promotes the activity of endogenous proteases [50]. Moreover, starter cultures like lactic acid bacteria and staphylococci, play a significant role in proteolysis during the ripening of dry-cured fermented sausages through the action of microbial enzymes, generating large amounts of peptides and free amino acids [51]. The drying-ageing stage is also reported to significantly increase the total free amino acid concentration in different varieties of dry fermented sausages [52].

The curing process increases the rate of proteolysis; however, the addition of sodium nitrite/nitrate decreases this content, regardless of the concentration.

## 4. Conclusions

This study gives new insights into the formation of nitroso compounds in dry-cured fermented sausages. Among the highlights, the addition of 80 ppm of sodium nitrite and nitrate was sufficient to decrease lipid peroxidation without the formation of nitrosamines. Moreover, this concentration was also sufficient to decrease the release of free iron. This is a major result since the iron released is involved in the formation of oxidation products. Furthermore, the results showed that the use of starter cultures and the addition of sodium nitrite/nitrate promote nitrosylation. The nitrosamine content, higher in the high NO_2_/NO_3_ condition, stressed the importance of investigating the addition of ascorbate in dry-cured fermented sausage formulations.

This study provided new knowledge on the mechanisms of nitroso compound formation and oxidation reactions in a dry-cured fermented sausage. The next steps will be to establish the reaction pathways involved in the formation of NOCs during the digestion of cured fermented meat. Indeed, the conditions of the digestive environment, such as low gastric pH, reducing conditions and oxygen pressure promote the formation of nitrite derivatives, such as nitric oxide and nitrosonium ions, which are involved in the formation of nitroso compounds.

## Figures and Tables

**Figure 1 foods-10-00852-f001:**
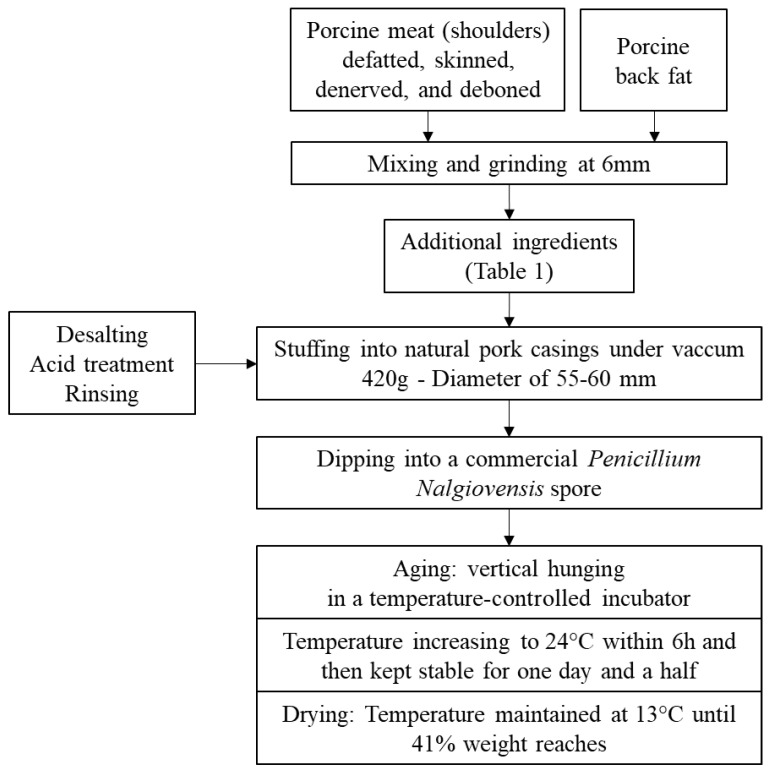
Elaboration process of dry-cured fermented sausages.

**Figure 2 foods-10-00852-f002:**
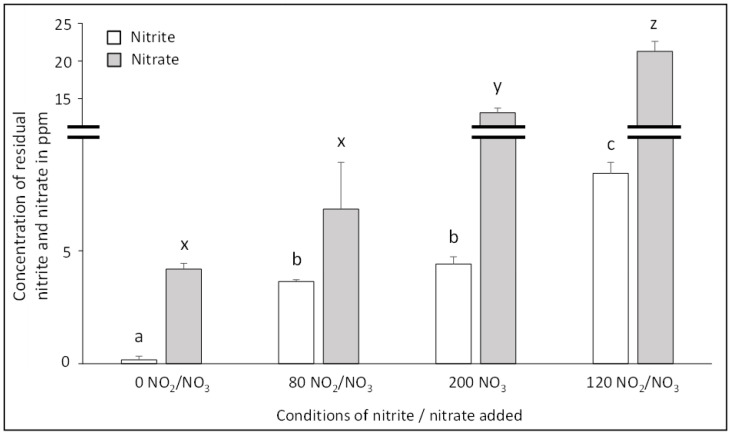
Effect of added doses of sodium nitrite and nitrate on the concentration of residual nitrite and residual nitrate in the cured and dry-cured fermented sausages. The rates of residual nitrite (in white) and residual nitrate (in grey) are expressed in ppm. Values are mean ± SEM of 6 independent determinations. Values without common superscripts, a, b, c for nitrite and x, y, z for nitrate, differ significantly (*p* < 0.01).

**Figure 3 foods-10-00852-f003:**
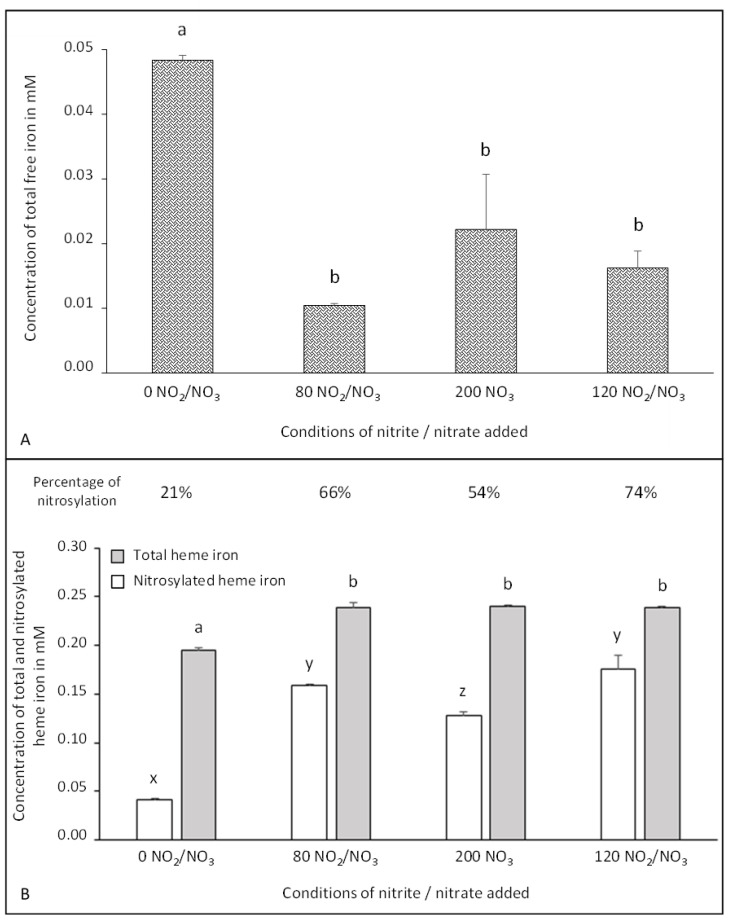
Effect of added doses of sodium nitrite and nitrate on the concentration of total free iron (**A**), nitrosylated heme iron and total heme iron (**B**) in dry-cured fermented sausages. The level of free iron, nitrosylated heme iron (in white) and total heme iron (in grey) are expressed in mM. Values are mean ± SEM of 6 independent determinations. Values without common superscripts, a, b for total free and heme iron and x, y, z for nitrosylated heme iron, differ significantly (*p* < 0.01).

**Figure 4 foods-10-00852-f004:**
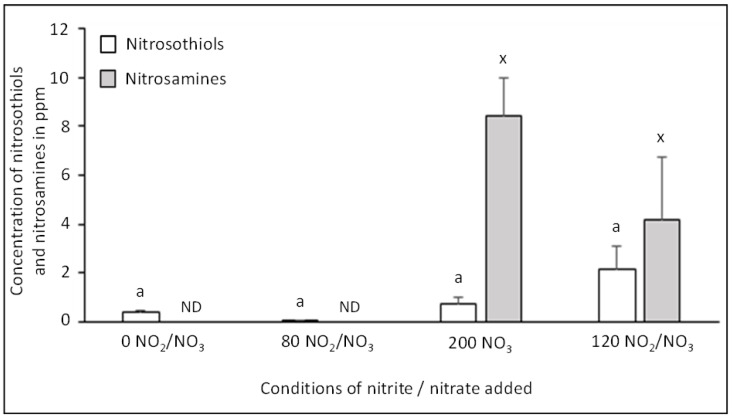
Effect of added doses of sodium nitrite and nitrate on the concentration of nitrosation products, nitrosothiols and nitrosamines, in dry-cured fermented sausages. The level of nitrosothiols (in white) and nitrosamines (in grey) are expressed in ppm Values are mean ± SEM of 6 independent determinations. Values without common superscripts, a for nitrosothiols and x for nitrosamines, differ significantly (*p* < 0.01). ND = not detected.

**Figure 5 foods-10-00852-f005:**
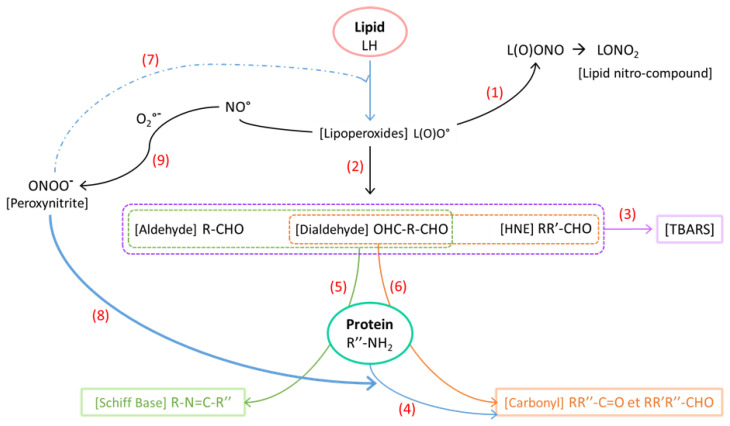
Reactions involved in the oxidation of lipids and proteins in dry-cured fermented sausages. The oxidation reactions are represented by blue arrows; reaction between nitric oxide and lipoperoxides: Reaction (1); formation of aldehyde from lipoperoxides: Reaction (2); formation of TBARS: Reaction (3); formation of carbonyl from protein oxidation: Reaction (4); formation of Schiff base: Reaction (5); formation of carbonyl from dialdehydes and HNE: Reaction (6); lipid oxidation by ONOO^−^: Reaction (7); protein oxidation by ONOO^−^: Reaction (8); formation of peroxynitrite (ONOO^−^): Reaction (9). Reaction (8) (in bold) is favored in comparison with Reaction (7) (in dashes) because peroxynitrite oxidizes preferentially proteins more than lipids.

**Figure 6 foods-10-00852-f006:**
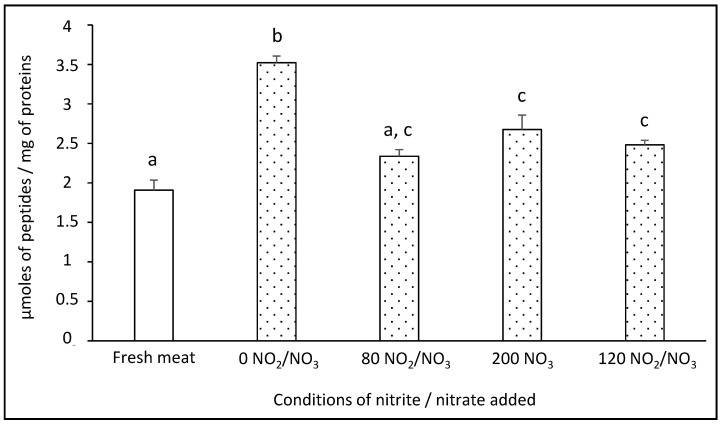
Effect of added doses of sodium nitrite and nitrate on the proteolysis index in dry-cured fermented sausages. The proteolysis index is expressed in µmoles of peptides/mg of proteins. Values are mean ± SEM of 6 independent determinations. Values without common superscripts, differ significantly (*p* < 0.01).

**Table 1 foods-10-00852-t001:** Composition of dry-cured fermented sausages (g/kg or ‰ of mixed meat-fat).

	0 NO_2_/NO_3_	80 NO_2_/NO_3_	200 NO_3_	120 NO_2_/NO_3_
Meat (shoulders)	870			
Back fat	130			
Ferment GY2 (*Lb. sakei, S. carnosus, S. xylosus*)	0.15			
Ground grey pepper	1.50			
Dextrose	5.50			
Lactose	6.00			
Natural salt (NaCl)	26.00	12.67	26.00	6.00
Nitrite salt (0.6% of NaNO_2_)	0.00	13.33	0.00	20.00
Potassium nitrate (KNO_3_)	0.00	0.10	0.24	0.14

**Table 2 foods-10-00852-t002:** Concentration in equivalent mM of total NO added in dry-cured fermented sausages.

	0 NO_2_/NO_3_	80 NO_2_/NO_3_	200 NO_3_	120 NO_2_/NO_3_
Dose NaNO_2_ (ppm)	0	80	0	120
Dose NaNO_3_ (ppm)	0	80	200	120
Concentration of total NO added (mM)	0	3	3.2	4.5

**Table 3 foods-10-00852-t003:** Effect of added doses of sodium nitrite and nitrate on the concentration of thiobarbituric acid reactive substances (TBARS), carbonyl and free thiol groups in dry-cured fermented sausages.

	0 NO_2_/NO_3_	80 NO_2_/NO_3_	200 NO_3_	120 NO_2_/NO_3_
TBARS µg MDA/g lipids	12.67 ^a^ ± 0.39	7.34 ^b^ ± 0.41	8.39 ^b^ ± 0.89	6.05 ^b^ ± 0.91
Carbonyl groups nmoles/mg proteins	2.44 ^a^ ± 0.06	2.74 ^b^ ± 0.04	2.79 ^b^ ± 0.12	2.79 ^b^ ± 0.06
Free thiol groups nmoles/mg proteins	33.72 ^a^ ± 1.87	32.65 ^a^ ± 3.38	30.68 ^a^ ± 2.12	26.26 ^a^ ± 1.66

The level of TBARS is expressed in µg MDA/g lipids and the level of carbonyl and free thiol groups are expressed in nmoles/mg of proteins. Values are mean ± SEM of 6 independent determinations. Values without common superscripts differ significantly (*p* < 0.01).

## Data Availability

The data presented in this study are available on request from the corresponding author.

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
