# Peer review of "New Insights into the Chemical Reactivity of Dry-Cured Fermented Sausages: Focus on Nitrosation, Nitrosylation and Oxidation"

_foods, 2021, doi:10.3390/foods10040852_

Round 1
Reviewer 1 Report
Title, please change ... fermented sausages for dry-cured fermented sausages. Also is not clear the use of cured meat, since experimental are only related to dry-cured fermented sausages. Can the authors explain this aspect?
Material and Methods: .... into natural casings, Please specify which type of assign was used (hog, bovine, sheep casings), diameter and previous conditioning.
As this experiment seems to be industrial please, change the formula in % for each ingredient.
For a better elaboration process comprehension, a flow chart would be very useful.
In-text, please use international units, please avoid the use of ppm.
Author Response
Dear editor,
We thank the reviewer for their comments and suggestions that have greatly improve the quality of our manuscript. All the comments were taken into account, please find below the answers.
Reviewers' comments:
Reviewer #1:
Title, please change ... fermented sausages for dry-cured fermented sausages.
Answer: The title has been changed according to the reviewer’s comment : “New insights into the chemical reactivity of dry-cured fermented sausages: focus on nitrosation, nitrosylation and oxidation”
Also is not clear the use of cured meat, since experimental are only related to dry-cured fermented sausages. Can the authors explain this aspect?
Answer: We thank the reviewer for bringing this inconsistency in vocabulary to our attention. The samples studied are now all referred to as "dry-cured fermented sausage" throughout the manuscript.
Material and Methods: .... into natural casings, Please specify which type of assign was used (hog, bovine, sheep casings), diameter and previous conditioning.
Answer: This part has been completed lines 108 - 109: “After being stuffed into natural pork casings with a diameter of between 55 and 60 mm and having previously undergone desalting, acid treatment and rinsing. Fresh sausages were dipped into a commercial Penicillium Nalgiovensis spore solution (BiovitecTex-el, Dupont Danisco, France) and hung vertically in a temperature-controlled incubator to carry on the ripening process”.
As this experiment seems to be industrial please, change the formula in % for each ingredient.
For a better elaboration process comprehension, a flow chart would be very useful.
Answer: A flow chart has been added: line 89 :” The elaboration process is summarized in Figure 1.”
In-text, please use international units, please avoid the use of ppm.
Answer: We added the concentration in nitrite, nitrate, nitrosothiol and nitrosamines,expressed in ppm in the article, in µM in supplementary material. We precised it on line 106: “Moreover, the concentrations of nitrite, nitrate, nitrosothiols and nitrosamines expressed in ppm are also expressed in µM in supplementary material (Table S1).”

Reviewer 2 Report
Comments to Authors:
In this study, the authors aimed to elucidate the reactions and interactions implicated in the formation of NOCs and the oxidation of lipids and proteins in cured and dry fermented meat products, with regard to adding doses of nitrite / nitrate. To this end, they investigated the effect of modulating added nitrite and/or nitrate concentrations in dry cured fermented sausages, in which the chemical reactivity of precursors and catalysts in the formation of nitroso compounds and lipid and protein oxidation products is evaluated.
The manuscript is well written in skillful English, and shows the appropriate data to demonstrate what they aimed to investigate, which would contribute to the scientific field of meat products. I recommend some minor revisions.
Line 63, 270, 302, etc.: In meat science, normally aging or conditioning is used for changing process during postmortem storage of meat, instead of ripening. Please reconsider the word usage.
Line 254: Please add references for description of this sentence.
Line 432: I did not find Figure 5. Need to add the figure.
Please make sure all the references.
Author Response
Dear editor,
We thank the reviewer for their comments and suggestions that have greatly improve the quality of our manuscript. All the comments were taken into account, please find below the answers.
Reviewers' comments:
Reviewer #2:
In this study, the authors aimed to elucidate the reactions and interactions implicated in the formation of NOCs and the oxidation of lipids and proteins in cured and dry fermented meat products, with regard to adding doses of nitrite / nitrate. To this end, they investigated the effect of modulating added nitrite and/or nitrate concentrations in dry cured fermented sausages, in which the chemical reactivity of precursors and catalysts in the formation of nitroso compounds and lipid and protein oxidation products is evaluated.
The manuscript is well written in skillful English, and shows the appropriate data to demonstrate what they aimed to investigate, which would contribute to the scientific field of meat products. I recommend some minor revisions.
Line 63, 270, 302, etc.: In meat science, normally aging or conditioning is used for changing process during postmortem storage of meat, instead of ripening. Please reconsider the word usage.
Answer: The word “ripening” has been replaced by “aging”thoughout the manuscript, lines 63, 270 and 302.
Line 254: Please add references for description of this sentence.
Answer: A reference has been added (now line 253): “thereby preventing the formation of H2O2 involved in heme iron release [34].”
Line 432: I did not find Figure 5. Need to add the figure.
Answer: Indeed, Figure 5 has been added.
Please make sure all the references.
Answer: References have been verified.

Reviewer 3 Report
(Title) New insights into the chemical reactivity of a cured and dry fermented meat product: focus on nitrosation, nitrosylation and oxidation
<Overall>
○ This study investigated the dose-response relationship of added sodium nitrite and/or sodium nitrate on the formation of nitroso compounds in dry cured fermented sausages. This paper covers appropriate topics that are of current interest. Although this manuscript is well written, some parts need more specification to ensure high quality.
<Specification>
Materials and Methods
- L88-93: In the case of porcine shoulders, 15-20% of fat is deposited, so it is difficult to separate the fat. Is there a reason for choosing a shoulder meat instead of other simple parts?
- L142: The 2 in HgCl2 should be written as a subscript.
Results and Discussion
- L249-253: References to the presented discussion are required.
- L432: There is no Figure 5 in this manuscript. Authors should insert Figure 5 into the manuscript.
Author Response
Dear editor,
We thank the reviewer for their comments and suggestions that have greatly improve the quality of our manuscript. All the comments were taken into account, please find below the answers.
Reviewers' comments:
Reviewer #3:
This study investigated the dose-response relationship of added sodium nitrite and/or sodium nitrate on the formation of nitroso compounds in dry cured fermented sausages. This paper covers appropriate topics that are of current interest. Although this manuscript is well written, some parts need more specification to ensure high quality.
Materials and Methods
L88-93: In the case of porcine shoulders, 15-20% of fat is deposited, so it is difficult to separate the fat. Is there a reason for choosing a shoulder meat instead of other simple parts?
Answer: The shoulder is rich in heme iron, the goal was to use a piece to promote the reactions we wanted to observe, including nitrosylation, to better study them.
L142: The 2 in HgCl2 should be written as a subscript.
Answer: It is corrected.
Results and Discussion
L249-253: References to the presented discussion are required.
Answer: A reference has been added (now line 253): “thereby preventing the formation of H2O2 involved in heme iron release [34].”
L432: There is no Figure 5 in this manuscript. Authors should insert Figure 5 into the manuscript.
Answer: Indeed, Figure 5 has been added.
